# Recommendations for Administrators’ Involvement in School-Based Health Promotion: A Scoping Review

**DOI:** 10.3390/ijerph17176249

**Published:** 2020-08-27

**Authors:** Collin A. Webster, Genee Glascoe, Chanta Moore, Brian Dauenhauer, Cate A. Egan, Laura B. Russ, Karie Orendorff, Cathy Buschmeier

**Affiliations:** 1Department of Physical Education, University of South Carolina, Columbia, SC 29208, USA; 2Department of Educational Studies, University of South Carolina, Columbia, SC 29208, USA; gglascoe@email.sc.edu (G.G.); cmoore2@email.sc.edu (C.M.); 3School of Sport and Exercise Science, University of Northern Colorado, Greeley, CO 80639, USA; brian.dauenhauer@unco.edu (B.D.); Cathybuschmeier15@hotmail.com (C.B.); 4Department of Movement Sciences, University of Idaho, Moscow, ID 83844, USA; eganca@uidaho.edu; 5Independent Researcher, Albany, GA 31701, USA; laurabruss@gmail.com; 6Department of Health and Human Development, Montana State University, Bozeman, MT 59717, USA; karie.orendorff@montana.edu

**Keywords:** wellness, physical activity, physical education, nutrition, principal, superintendent, educational leadership

## Abstract

School administrator involvement is recognized as a key factor in the extent to which school health promotion programs and initiatives are successfully implemented. The aims of this scoping review are to: (a) Identify existing documents that contain recommendations regarding the involvement of school administrators in school-based health promotion; (b) distill and summarize the recommendations; (c) examine differences in the recommendations by targeted professional level, professional group, health promotion content focus, and by whether the recommendations are evidence-based or opinion-based; and (d) evaluate the research informing the recommendations. We drew upon the Preferred Reporting Items for Systematic Reviews and Meta-Analyses extension for Scoping Reviews (PRISMA-ScR) guidelines to conduct the review. Our team conducted a comprehensive literature search with no date or geographic restrictions from January 2018 through April 2018 using four electronic databases: Academic Search Complete, Google Scholar, Physical Education Index, and PubMed. Eligibility criteria included any online documents, in English, that contained recommendations targeting school administrators’ (e.g., principals, assistant principals, superintendents) involvement (e.g., support, endorsement, advocacy) in school health programming (e.g., physical activity, nutrition, wellness). The search yielded a total of 1225 records, which we screened by title, then by abstract, and finally by full text, resulting in 61 records that met inclusion criteria. Data (e.g., recommendations, targeted contexts, targeted administrators) from these records were extracted for a content analysis. Included records contained 80 distinct recommendations, which we summarized into three themes (Collaboration, Advocacy, and Support) using a content analysis. Separate content analyses revealed no qualitative differences in the recommendations by professional level, professional group, or content focus, or by whether the recommendations were evidence-based or opinion-based. Twenty-one of the included records were peer-reviewed research articles. Using the Mixed Methods Appraisal Tool (MMAT), we appraised qualitative research articles the highest and mixed methods research articles the lowest. This review provides a basis for future research and professional practice aiming to increase school administrators’ involvement in school-based health promotion.

## 1. Introduction

For decades, it has been well understood that schools should play an important and comprehensive role in promoting the health of children and adolescents [1]. Good health and educational attainment are not mutually exclusive for school-age youth: addressing students’ health disparities is critical to closing the academic achievement gap [2]. Moreover, characteristics of effective schools, such as the coordination of efforts among multiple stakeholders (e.g., teachers, administrators, families), overlap with characteristics of effective school health programs [3]. Thus, clear links exist in terms of both process and outcome between the tandem pursuits of educational effectiveness and a health promoting environment in schools.

Internationally, numerous school health interventions have demonstrated effectiveness in enhancing students’ academic- and health-related outcomes [4]. However, the uptake and sustainability of school health programs and initiatives have remained low [5]. A key factor in the extent to which schools implement educational initiatives and innovations, including evidence-based health programs, is the support of school administrators [6,7,8]. Administrators, such as assistant principals, principals, and district officials, enact their support for school policy and programming through various forms of involvement. For example, Durand et al. [6] found that district leaders for schools that performed better than expected on state standards developed organizational readiness and capacity building strategies that supported the successful implementation of the standards. With respect to school health interventions, Storey et al. [9] found that supportive principal involvement for the implementation of a comprehensive school health approach included prioritizing the initiative in the school’s agenda, actively serving on the implementation team, and being an advocate and a role model for the initiative.

The conceptual and theoretical bases for focusing on administrator involvement in efforts to promote school heath are firmly rooted in the Whole School, Whole Community, Whole Child (WSCC) model [10]. Bridging perspectives from education and public health, the WSCC model outlines a systems-based approach to supporting students’ learning and development through health promotion. School administrators are identified as essential to the coordination and collaboration needed across ten health promoting components of the school: (a) health education, (b) physical education and physical activity, (c) nutrition environment and services, (d) health services, (e) counseling, psychological, and social services, (f) social and emotional climate, (g) physical environment, (h) employee wellness, (i) family engagement, and (j) community involvement. It is proposed that for school health promotion to be effective, administrators must be involved with the work of school and district wellness teams and in “coordinating policy, process, and practice and integration of the community” (p. 736).

Additional specification of administrator involvement and its impact on school health promotion can be found in relation to specific components of the WSCC model. For example, Carson et al.’s [11] conceptual framework for research and practice related to comprehensive school physical activity programs (CSPAPs) addresses the physical education and physical activity component of the model. Consistent with the model’s systems-based approach, Carson et al. place administrator support within a social-ecological frame to illustrate the interconnectedness of different factors on students’ physical activity. Administrator involvement is proposed to consist of multiple forms and examples are given in terms of emotional support (e.g., praising staff and students who help to promote physical activity), instrumental support (e.g., providing resources to staff for physical activity promotion), and informational support (e.g., offering feedback to staff on their efforts to promote physical activity). In coordination with a CSPAP committee and a program champion, school administrators are theorized to directly influence the CSPAP facilitators, including the knowledge, skills, dispositions, resources, and safety needed to implement the program.

Given the important function of administrators in the uptake of programs and initiatives in school settings, and based on the above-mentioned conceptual and theoretical propositions concerning administrator involvement in school health promotion, research aimed at further elucidating the array of recommended roles, responsibilities, and behaviors that comprise such involvement is warranted. To the authors’ knowledge, no study to date has documented the corpus of literature on administrator involvement in school health promotion or attempted to reduce this literature into a distinct set of recommendations that can underpin operational definitions of administrator involvement in future research. Operationally defining administrator involvement can help to advance both research and practice related to school-based health promotion. Specifically, it can lead to the development of appropriate instruments for measuring administrator involvement in the context of descriptive research and in intervention studies designed to enhance administrator involvement. Having a comprehensive set of clear recommendations for administrator involvement can also support pre-professional and in-service learning opportunities for school administrators. These benefits to research and practice can ultimately facilitate the implementation and sustainability of school health programs and initiatives.

One approach for identifying, collating, and synthesizing targeted information from existing literature is to conduct a scoping review. According to Mays et al. [12], scoping reviews canvas “concepts underpinning a research area and the main sources and types of evidence available, and can be undertaken as stand-alone projects in their own right, especially where an area is complex or has not been reviewed comprehensively” (p. 194). Researchers perform scoping reviews to meet several different objectives, which include (a) examining the extent, range, or nature of the evidence on a topic, (b) determining whether there is value in subsequently conducting a systematic review, identifying gaps in the literature to plan for future research, or (d) summarizing a body of knowledge across heterogenous sources or methods [13]. The present study was conducted in line with this last objective. Our aim was to first identify available online documents that contained recommendations pertaining to the involvement of school administrators in school-based health promotion, and then to summarize the recommendations and evaluate the research informing the recommendations. The specific research questions driving this review are:What types of documents contain recommendations pertaining to administrators’ involvement in school health promotion?What are the distinct recommendations and their summarized themes, across all included documents?Are there qualitative differences between recommendations in terms of their targeted level of professional practice (P-12 schools versus higher education), targeted professional group (e.g., principals, district superintendents, administrator preparation programs), health promotion content focus (e.g., physical activity, nutrition, wellness) or whether they are evidence-based versus opinion-based?What is the quality of the existing research studies included in the review?

## 2. Methods

The Preferred Reporting Items for Systematic Reviews and Meta-Analyses extension for Scoping Reviews (PRISMA-ScR) guidelines [13] informed this review.

### 2.1. Inclusion and Exclusion Criteria

We restricted our literature search to documents available via the Internet, including the University of South Carolina library’s online databases, which the first and third author (who conducted the search) could access. In line with recommendations for conducting scoping reviews [14], we used PCC (Population, Concept, and Context) as a framework to determine our inclusion and criteria (Table 1). The target population was school administrators, the target concept was involvement in school-based health promotion, and the target contexts were P-12 schools and higher education. Therefore, the inclusion criteria we used for records to be included in the review were as follows: (a) Contained recommendations or guidelines for school administrators (e.g., assistant principals, principals, superintendents) with respect to their role in promoting health-related programing within the school community (e.g., on and off campus, with students, staff or parents); or (b) contained recommendations or guidelines for preservice school administrator preparation programs with respect to preparing future school administrators for involvement in (e.g., to be advocates, supporters, and/or promoters of) health-related programming within the school community; or (c) contained recommendations or guidelines for in-service professional development with respect to training school administrators to be involved in health-related programming within the school community. We excluded documents if they were not available in full online, did not meet at least one of the inclusion criteria or were not in English.

### 2.2. Search Strategy

We conducted a search from January 2018 through April 2018 for relevant documents, utilizing four electronic catalog databases: Academic Search Complete, Google Scholar, Physical Education Index, and PubMed. Prior to conducing a formal search, we used Google Scholar (sorting results by relevance) to do some preliminary searches of the literature, in which we examined up to the first 100 returned records of each search. This helped us to determine the most relevant search terms, which we categorized into four dimensions (Table 2). We did not place date restrictions on our searches and considered documents from any country as long as they were written in English. One researcher (the third author) independently completed the search following several practice trials with the first author to become familiar with the search strategy and databases. The search encompassed peer-reviewed articles in academic or professional journals, unpublished theses or dissertations, conference proceedings, published abstracts, books, book chapters, position statements, professional guidance documents, and policy briefs.

### 2.3. Screening Process

Figure 1 displays the steps taken to reduce the full set of records yielded from the searches to a final set of records included for data analysis. First, one researcher (the second author) removed duplicates from the list of all records by title. Subsequently, two researchers (the second and last authors) excluded records with titles that were clearly irrelevant to the study’s purpose, then excluded additional records based on the information provided in abstracts, where applicable. These researchers then obtained and screened the full texts of the remaining records to identify the final set of documents (*N* = 61) for inclusion in the data analysis. In cases of uncertainty or disagreement about whether a record was eligible for the review, a third researcher (the first author) was consulted and he made the decision to include or exclude the document.

### 2.4. Identification of Document Types and Data Extraction

After screening records for inclusion, we addressed the first research question by identifying the type of document each record represents (e.g., peer-reviewed article, book chapter, position statement) and conducting a frequency count for each type of document identified. Subsequently, in preparation for addressing the second and third research questions, we completed data extraction from all included records. This involved reading each document and identifying text (e.g., phrases, sentences, passages) that focused on one or more recommended roles or responsibilities of any school administrator (e.g., principal, assistant principal, district superintendent) with respect to any school-based health program or initiative (e.g., physical education, recess, nutrition program, staff wellness program). We copied relevant text from each record and pasted into a separate file, then treated the text as data for this study. Two researchers (the first and fourth authors) confirmed that all relevant text from the included records was captured for analysis.

### 2.5. Data Analysis and Quality Appraisal of Research Studies

Two of the researchers (the first and fourth authors) independently read through and summarized the extracted text to distill the embedded recommendations. For each recommendation, the researchers identified the following characteristics: (a) target level (i.e., in-service or preservice), (b) target group (e.g., assistant principals, principals, district superintendents), and (c) content focus (e.g., physical activity, physical fitness, nutrition, wellness). Additionally, another researcher (the second author) determined whether the recommendations were evidence-based or opinion-based. For the purposes of this review, we considered recommendations to be evidence-based if they were contained in the discussion section of a peer-reviewed research article or systematic review. The fifth and sixth authors verified the assignment of recommendations as evidence- or opinion-based.

The first and fourth authors then discussed their work, came to agreement on the text summaries (i.e., distilled recommendations) and characteristics (i.e., target level, target group, content focus, and evidence-/opinion-based), and together developed a color-coding scheme to categorize the recommendations according to their core emphases. For example, the researchers created a category called Professional Development and Technical Assistance to collate recommendations that emphasized the importance of administrators providing workshops and trainings for teachers and others who can play a role in school-based health promotion. The researchers independently tested the coding scheme with approximately one-third of the recommendations and then met again to discuss the evolving recommendation categories, finding few instances of disagreement, which the researchers discussed until reaching consensus. Subsequently, the fifth and sixth authors reviewed this coding process, confirmed its dependability, and proceeded to categorize the remaining recommendations.

To address the second research question, the fifth and sixth authors summarized the recommendation categories by grouping them into broader recommendation themes. A qualitative content analysis was used to develop the themes [15]. Specifically, we employed an inductive process whereby we compared the recommendation categories in search of commonalities in their latent meanings. As we developed the themes, we used negative case analysis to test whether they were accounting for all recommendation categories [16]. This involved searching for recommendation categories that did not seem to align with any of the themes and continually revising the themes until no category was unaccounted for. To address the third research question, the same two authors conducted separate content analyses of the recommendations by their targeted level of professional practice (P-12 schools vs. higher education), targeted professional group (assistant principals, principals, etc.), health promotion content focus (physical education, wellness, etc.), and whether they were evidence-based or opinion-based. To address the fourth research question, the first and fifth authors independently appraised all records identified as peer-reviewed research articles using the Mixed Methods Appraisal Tool (MMAT) Version 2018 [17]. These authors then met to compare their evaluations, discuss any disagreements, and reach consensus for each article.

## 3. Results

### 3.1. Research Question 1: Types of Documents Identified in Included Records

Across the 61 included records in this review, we identified six different types of documents (Table 3). Peer reviewed articles in academic and professional journals constituted the largest proportion (88.5%) of the included records, with the majority of articles identified as either professional journal articles (i.e., articles written primarily for practitioners, such as school staff) or research studies. Other types of documents included two unpublished doctoral dissertations, one book, one book chapter, one professional guidance document, and one published abstract for a research study.

### 3.2. Research Question 2: Distinct Recommendations and Their Summarized Themes

We identified a total of 80 distinct recommendations from the extracted text of the included records and grouped the recommendations into 10 different categories (Table 4). Across these categories, three themes were apparent and we labeled the themes Collaboration, Advocacy, and Support.

#### 3.2.1. Theme 1: Collaboration

The theme, Collaboration, incorporated the most recommendations (44) of the three themes. Recommendations within this theme focused on the importance of administrators supporting collaboration between key stakeholders (e.g., teachers, school boards, wellness teams, district officials, community partners, students) and building strong relationships among these stakeholders. There were three categories within this theme: Joint Effort, Implementation, and Involvement in Planning and Programming. Joint Effort was based on recommendations for administrators to work with teachers and other stakeholders on developing facility joint-use agreements [18] and allowing community members to access school facilities and equipment [19]; work with teachers and board members to promote health and improve access [20]; use departmental/staff meetings to gather input for establishing school wide goals [21]; build a strong and sustainable relationship inside the school and with community stakeholders [22]; develop rapport with the community and ensure the community feels a sense of ownership for the school [22]; share data with students, staff and parents and empower these groups to use the data for decision-making [22]; work with all stakeholders to foster positive relationships [21,23], promote healthy school environments [20], provide resources [24,25], improve access to and ensure the quality of health-related school programming [20,24,25], and ensure each person/group is achieving their goals [26,27]; and work with stakeholders to organize special events (e.g., fundraisers, Walk to School Days) [28].

Embedded within the second category, Implementation, were recommendations for administrators to lead or co-lead initiatives [22]; encourage and foster enthusiasm for health promotion at school [29]; use a distributive leadership style (e.g., transition progressive and caring faculty to leadership positions) [22]; embrace resistance [22]; communicate effectively and make sure that teachers across the school understand the program vision and its role in school improvement [22]; engage in a high level of communication with other stakeholders during the implementation process [30]; use systems to monitor implementation [31]; assist in the implementation of evidence-based practices for school programming (e.g., physical education) [32,33]; schedule school wide physical activity sessions each day [29]; implement a school intramural program and encourage staff involvement [34]; implement physical activity in classrooms and encourage classroom teacher involvement [35,36]; go on playground supervision cycles, be visible to students and interact with them at recess and during other physical activity opportunities [34]; encourage teachers to be creative, implement new ideas, and customize health-related programming to the school context [37]; adjust teacher workload to facilitate implementation, integrate PA opportunities into the school curriculum, and acknowledge teachers’ health promotion efforts [38]; work in tandem with other administrative staff and teachers to implement programming [39]; and liaise between school and community resources to enhance implementation [40,41].

Recommendations encapsulated in the third category, Involvement in Planning and Evaluation, included having administrators engage in formal planning for health promotion [42]; incorporate health-related goals into the school’s vision and mission statements and/or school improvement plans [43]; plan to address challenges and barriers [43]; actively seek ways to increase physical activity time beyond physical education and recess while keeping the academic schedule intact [29]; use student data in health promotion planning and engage in a high level of communication with other stakeholders during the planning process [30]; collaboratively build a broad conception of school health programming (i.e., beyond physical education) [23]; consider all resources (e.g., student interest, school resources, community resources) at their disposal [44]; contribute to the overall vision of the school’s health promotion plan [45]; be involved in defining the health problems, identifying solutions, and evaluating progress [46]; identify and use assessment tools that include specific behaviors related to health promotion [47]; audit lesson plans and provide feedback to ensure incorporation of targeted health promotion strategies [29]; and serve on a school wellness committee [48].

#### 3.2.2. Theme 2: Advocacy

The second largest theme, Advocacy, included 29 recommendations that centered on the idea that administrators should value health promotion and its position within the school, gain more experience related to health promotion, and advocate for health promotion policies and the expansion of relevant knowledge to others in the school community. Accordingly, Advocacy consisted of three categories: Knowledge and Values, Policy Involvement, and Exposure and Experience. The largest category, Knowledge and Values, included recommendations for administrators to become informed about physical activity as a learned behavior and about what a quality physical education program is [24,35]; make a commitment to, and prioritize, student health and wellness [26,29,49,50]; understand and appreciate that health promotion has to be a school wide effort [51]; believe in the importance of hiring well-trained physical education teachers [52]; recognize and value the importance of health and wellness (e.g., physical activity) [35,41]; view monitoring children’s health and wellbeing as part of their responsibility [25]; view health and wellness as inextricably linked to other school priorities [25,53]; know what health promotion strategies are effective with students [54]; and display a belief in faculty, staff and team members with respect to health promotion [22]. At the preservice level, authors recommended that health-promoting practices and health promoting models/frameworks (e.g., comprehensive school physical activity program model) be embedded into administrator preparation programs [55,56]. In addition, Illg [57] suggested that teacher educators help preservice administrators develop an understanding of the benefits of physical activity, how to implement school-based health programming, and how such programming can positively influence the school community.

In the second category, Policy Involvement, recommendations focused on administrator involvement with policies and accountability for school-based health promotion. Specifically, authors recommended that administrators endorse or develop policies for health promotion [31,42,58]; educate policymakers about the importance of school health programs (e.g., physical education) [59]; hold teachers and other school staff accountable for implementing health promotion initiatives [60]; consider integrating standardized practices, roles, and responsibilities for implementation teams as part of policy development [56]; and constantly pursue sustained change in school structures, effective practices, and sound policies [22].

The third category, Exposure and Experience, consisted of recommendations for administrators to gain exposure and experience related to school-based health promotion and also to increase exposure of school-based health programming to other stakeholders. In terms of increasing administrators’ own exposure and experience, recommendations were for administrators to attend and participate in professional development trainings and workshops related to school-based health promotion, including those designed for teachers [56,61]. Recommendations for administrators to increase exposure of the programs at their schools were to increase overall recognition/awareness of school health programming [24]; advocate for a whole-school approach to health promotion [59]; role model healthy behaviors [40]; and maintain a visible presence [22].

#### 3.2.3. Theme 3: Support

The final theme, Support, was comprised of four recommendation categories, including Funding/Resources, Professional Development/Technical Assistance, Prioritization of Physical Education, and General Support. These categories encompassed a total of 13 recommendations that focused on different ways school administrators should provide support for school-based health promotion. The Funding/Resources category was based on recommendations for administrators to allocate funding to health promotion programs (e.g., physical education) [42,43,62]; require, fund and allocate personnel to lead health promotion initiatives [42,43,56]; offer incentives for PA promotion [62]; provide materials to teachers for classroom-based PA [61]; provide resources to all teachers for supporting school-based health programming (e.g., quality physical education) [22,24]; and capitalize on free resources, such as America on the Move (www.americaonthemove.org), MyFitnessPal (www.myfitnesspal.com), and EveryMove (www.everymove.org) [63].

Recommendations related to Professional Development/Technical Assistance included providing ongoing professional development, including specialized training for physical education teachers [22,48,61,64]; providing specific trainings for classroom teachers [61,65]; visiting teachers’ classrooms and offering feedback [61]; and providing workshops to a range of stakeholders on the benefits of physical education [24,66].

Closely aligned with the focus on funding/resources, recommendations within the category, Prioritization of Physical Education, had to do with ensuring administrators make physical education a viable part of the school curriculum [34,67] and allocate nationally recommended curriculum time for physical education in the school schedule [41,45]. As identified in Hickson et al. [68], “School leaders can assist health and physical education programming by ensuring that these subject areas figure prominently in a child’s weekly timetable, if not daily schedule” (p. 11).

The final category, General Support, embodied a single distinct recommendation that focused on the general importance of administrator support. Across numerous documents, we identified recommendations indicating that administrator support is important to school-based health promotion but these recommendations did not specify what such support should entail [19,20,22,24,25,31,37,39,68,69,70,71,72,73,74,75,76,77,78]. For instance, Centeio et al. [70] asserted, “the support of a local administrator is a necessity. Without support from a school administrator teachers may encounter barriers that prevent successful implementation” (p. 506).

### 3.3. Research Question 3: Differences in Recommendations by Target Level, Target Group, Content Focus, and Evidence-Based versus Opinion-Based

Table 5 presents the breakdown of the 80 distinct recommendations, based on the target level, target group, and content focus of the recommendations, and whether the recommendations are evidence-based versus opinion-based. All but three of the recommendations focused on administrator involvement in the P-12 school setting as opposed to what university programs might do to prepare future administrators to be involved in school-based health promotion. Additionally, nearly all of the recommendations were partially or fully directed at unspecified professional groups (e.g., “administrators,” “school leadership,” “school officials,” “policymakers”). Where recommendations did target specified groups, school principals were targeted the most frequently (48.8%) and ambiguity remained for exactly who recommendations targeting “district personnel” or the “State Department of Education” were written for. The focus of health promotion content in the recommendations largely centered on physical activity (93.8%), followed by approximately one-third of the recommendations focusing fully or in part on health, and about one quarter of the recommendations focusing fully or in part on physical education. Across all recommendations, over two-thirds (70%) were classified as evidence-based.

Content analyses revealed no qualitative differences between recommendations targeting different professional levels, professional groups, or health promotion content. Specifically, no unique themes were identified when separately considering recommendations that focused on the context of P-12 schools (e.g., principals’ involvement in school health promotion) versus higher education (e.g., strategies administrator education programs can use to prepare future school administrators to be involved in school health promotion); the involvement of principals versus district officials or other administrators; or the specific roles and responsibilities of administrators in regard to different areas/components of school health promotion (e.g., physical education, recess, nutrition programming, staff wellness). Further, qualitative differences were not evident from the content analyzing recommendations that were evidence-based versus opinion-based.

### 3.4. Research Question 4: Quality Appraisal of Research Articles

Twenty-one of the 61 included records in this review were peer-reviewed research articles. The results of our quality appraisal of these articles are presented in Table 6. The first step of the appraisal for all research articles is to answer two screening questions: “Are there clear research questions?” and “Do the collected data allow for the research questions to be addressed?” Response options include “yes,” “no,” and “can’t tell.” For each article, our response to these questions was “yes.” The remaining questions are specific to each of five study designs and use the same response options. One study design (quantitative—randomized controlled trial) was not used in any of the articles and is therefore omitted from Table 6.

Overall, qualitative research articles were appraised the highest in comparison to articles identified as having other research designs, whereas mixed methods research articles were appraised the lowest. Items with the lowest appraisals for each research design were “Is the interpretation of results sufficiently substantiated by data?” (qualitative), “Are there complete outcome data?” (quantitative—nonrandomized), “Is the risk of nonresponse bias low?” (quantitative—descriptive), “Are divergences and inconsistencies between quantitative and qualitative results adequately addressed?” (mixed methods), and “Do the different components of the study adhere to the quality criteria of each tradition of the methods involved?” (mixed methods).

## 4. Discussion

Despite the critical importance of administrator involvement for school-based health initiatives to succeed [9], little research has considered the range and variety of duties and responsibilities such involvement might entail. To our knowledge, this is the first scoping review of the literature on administrator involvement in school-based health promotion. The aims of this review were to (a) identify existing documents that contain recommendations regarding the involvement of school administrators in school-based health promotion, (b) distill and summarize the recommendations, (c) examine differences in the recommendations by targeted professional level, professional group, health promotion content focus, and by whether or not the recommendations are evidence-based or opinion-based, and (d) evaluate the research informing the recommendations.

### 4.1. Types of Documents Containing Recommendations

The vast majority of identified documents included in this review were either peer-reviewed professional articles or peer-reviewed research articles. Although the research articles date back to 1999, most of them were published in the past decade, indicating a more recent interest in considering the role of school administrators in school-based health promotion research. A trend of more recent published and unpublished literature was also evident for a large proportion of the remaining peer-reviewed articles (e.g., systematic review, literature/narrative reviews, commentaries, professional articles), as well as other types of documents (e.g., book chapter, unpublished doctoral dissertations, published abstract). The recently increased interest in administrator involvement underscores the importance of this scoping review, which can serve as a foundation for continued investigation into specific types of administrator involvement and can help to frame administrator preparation efforts for both current and aspiring school administrators.

### 4.2. Themes in the Recommendations

Based on the content analysis of the recommendation categories across all documents included in this review, there are myriad ways through the themes of collaboration, advocacy, and support, in which administrators might be involved in promoting health at their schools or within their school districts. These themes and their subthemes (e.g., joint effort, policy involvement, professional development/technical assistance) closely align with the conceptual and theoretical bases found in the WSCC model [10] and in frameworks applied to specific components of the model (e.g., CSPAP framework [11] for addressing physical education and physical activity). The WSCC model emphasizes the need for school administrators to work in coordination with school and district wellness teams to promote school health through “policy, practice, process, and integration of the community” (p. 736). Moreover, in describing their conceptual model to support CSPAP research and practice, Carson et al. [11] identify multiple forms of administrator involvement from a social-ecological perspective, including emotional, instrumental, and informational support. This review builds on these conceptual and theoretical perspectives by further defining the specific ways in which administrators might engage in school-based health promotion. As most recommendations for administrator involvement broadly targeted physical activity and health, with more specific recommendations focusing in large part on physical education programs, this review may particularly be useful in informing efforts targeting administrator involvement in these areas of health promotion. However, the recommendations analyzed in this review mainly neglected other components of the WSCC model, such as counseling, psychological, and social services; the social and emotional climate; and employee wellness, suggesting that there is a need for increased attention to administrator involvement in these additional areas of school health.

Almost all of the recommendations focused on the P-12 school context and in-service (currently practicing) school administrators, with only three recommendations targeting the higher education context (e.g., preservice administrator preparation programs). Nevertheless, consistent with Reynolds’ [79] perspective on applying research to practice in the preparation of school leaders, we believe the three recommendations themes identified in this study can serve as a guiding framework at both the in-service and preservice levels. Preservice program curricula, in-service workshops, and intervention programs can align professional learning opportunities with these recommendations themes to help aspiring and current school administrators learn how to leverage their leadership roles amid efforts to promote health within schools. In future studies, researchers could use these themes as a conceptual framework to develop appropriate measures for examining the extent to which administrators possess knowledge, enact skills, or receive training consistent with the recommendations.

### 4.3. Differences in the Recommendations by Targeted Professional Level, Targeted Professional Group, and Health Promotion Content Focus, and by Evidence-Based versus Opinion-Based

Qualitative differences were not apparent in this study regarding the focus of recommendations targeting different professional levels (P-12 schools versus higher education), groups of administrators (principals, district superintendents, etc.,), or health promotion content (physical activity, nutrition, etc.,). The recommendations provide little guidance in terms of how administrator involvement might look different for subsets of administrators (e.g., principals, assistant principals, district officials), despite distinct role articulations in the literature [80]. Additionally, based on existing recommendations, it is unclear how administrator involvement might change depending on the focus of a program/initiative (e.g., physical activity, nutrition, wellness). It will be important in future studies to determine which types of involvement by different administrative groups are most highly predictive of successful implementation and long-term continuance of school-based health initiatives. Furthermore, based on these studies, we advise authors to make recommendations that target specific groups of administrators so that clearer guidance about administrator involvement can be ascertained. Distinctions were also absent when comparing evidence-based versus opinion-based recommendations in this study. We therefore suggest that there is, as yet, no reason to give more attention or priority to evidence-based recommendations compared to opinion-based recommendations, as both evidence- and opinion-based recommendations focused on the same administrator behaviors captured in the themes (collaboration, advocacy, and support).

### 4.4. Quality Appraisal of Research Articles

The results of the quality appraisal of the 21 research articles included in this review indicated that qualitative research informing the recommendations for school administrator involvement in school-based health promotion tends to be of a higher quality, overall, than quantitative or mixed-methods research. It is important, however, for more authors of qualitative studies to clearly link findings to data sources to increase the credibility of their research [81]. Quantitative research lacked any randomized controlled trials, which are considered the gold standard for addressing questions about the effectiveness of interventions [82] and should be pursued to strengthen the evidence base informing recommendations for administrator involvement in school health. Our appraisal of mixed methods research articles revealed comparatively more issues than were found for qualitative or quantitative research articles. In many instances, authors were careful to demonstrate areas of convergence between qualitative and quantitative results but failed to address areas of inconsistency, which are equally important to consider, as such inconsistencies can help to guide future inquiry [83]. Authors of mixed methods articles also frequently did not take full advantage of both the qualitative and quantitative components in their research. For example, authors reported using new surveys but did not include any evidence of instrument validity or reliability to collect quantitative data. For the qualitative component of mixed methods studies, typical problems were that only one data source was used or the steps taken to analyze the data and ensure trustworthiness appeared to be insufficient. More robust mixed methods studies are needed so their potential contribution to the evidence base can be fully realized.

## 5. Conclusions

This review is significant because it is one of the first to foreground the role of administrators in the ongoing research related to school-based health promotion. While it is possible that we may have overlooked some relevant records in our search, we feel the findings are robust as they are based on 80 distinct recommendations across 61 documents. By collating and synthesizing extant literature to illuminate key recommendations for administrator involvement, we have uniquely distilled and articulated knowledge, skills, and dispositions that have repeatedly surfaced in the literature as important for school administrators to acquire and enact. However, this review does not provide an answer to the question about which of the recommendations are most likely to lead to heightened levels of administrator involvement. Increased research, particularly using rigorous quantitative designs and sound methods, is essential to advancing the science behind the recommendations and establishing an evidence base that can play a distinct and prominent role in guiding future practice. Overall, this review can underpin dedicated lines of inquiry on administrator training, preparedness, perceptions, and beliefs related to school-based health promotion.

## Figures and Tables

**Figure 1 ijerph-17-06249-f001:**
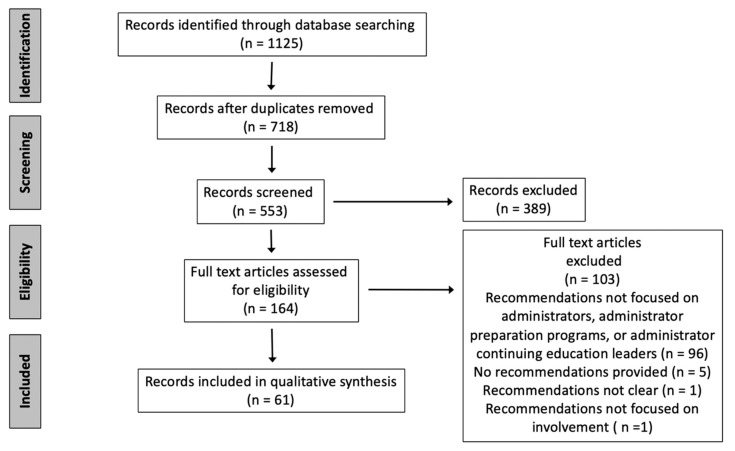
PRISMA flow diagram illustrating the process used to identify the final set of records included for qualitative synthesis.

**Table 1 ijerph-17-06249-t001:** PCC (population, concept, context).

P	School Administrators	Principals; assistant principals; district superintendents
C	Involvement in School-Based Health Programming	Advocacy, support, promotion or other kind of involvement in programs or initiatives related to school health (e.g., physical education, recess, wellness programs, nutrition initiatives), occurring before, during or after school, and targeting members of the school community (e.g., students, parents, staff)
C	P-12 Schools	Any context within the school community where school-based health programs or initiatives could reach (e.g., academic programs, extracurricular programs, students’ homes, community settings)
	Higher Education	Academic programs at colleges and universities that prepare students in the professional roles and responsibilities of school administrators
Other	Any full-text documents available online; English language only	

**Table 2 ijerph-17-06249-t002:** Search terms and their dimensions.

Dimension 1: Administrator’s Professional Role	Dimension 2: Involvement Behaviors	Dimension 3: Program/Initiative Focus	Dimension 4: Recommendations Context
“School leader” OR “school administrator” OR “school principal” OR “assistant principal” OR “vice principal” OR superintendent OR district official	Endorse OR support or advocate OR encourage OR allow OR permit	Wellness OR health OR fitness OR nutrition OR “physical activity” OR “physical education” OR recess OR CSPAP OR “Whole School, Whole Community, Whole Child”	Training OR “continuing education” OR “professional development” OR “teacher education” OR “preservice education” OR “graduate education”

**Table 3 ijerph-17-06249-t003:** Types of documents identified in the included records.

Document Type	Total Number	Percent
Peer-reviewed Article	55	90.2
Professional Journal Article	23	37.7
Research Study	21	34.4
Commentary/Other	5	8.2
Narrative/Literature Review	3	4.9
Mongraph Conclusion Chapter	1	1.6
Rationale/Study Design Paper	1	1.6
Systematic Review	1	1.6
Unpublished Doctoral Dissertation	2	3.3
Book	1	1.6
Book Chapter	1	1.6
Professional Guidance Document	1	1.6
Published Abstract	1	1.6

**Table 4 ijerph-17-06249-t004:** Themes, recommendation categories, and number of distinct recommendations in each category.

Theme	Recommendation Category	Total Number of Distinct Recommendations
Collaboration	Joint Effort	17
	Implementation	17
	Involvement in Planning and Evaluation	10
Advocacy	Knowledge and Values	14
	Policy Involvement	5
	Exposure and Experience	4
Support	Funding/Resources	6
	Professional Development/Technical Assistance	4
	Prioritizing Physical Education	2
	General Support	1

**Table 5 ijerph-17-06249-t005:** Breakdown of the recommendations based on their target level, target group, content focus, and whether they are evidence-based or opinion-based.

		Total Number	Percent
Targeted Professional Level	P-12 Schools	77	96.2
	Higher Education	3	3.8
Targeted Professional Group	Unspecified	75	93.8
	Prinicpals/School Managers	39	48.8
	District Personnel	11	13.8
	District Superintendents	1	1.3
	State Department of Education	1	1.3
Health Promotion Content Focus	Physical Acivity	75	93.8
	Health	25	31.3
	Physical Education	21	26.3
	Nutrition	6	7.5
	Wellness	6	7.5
Evidence-Based		56	70
Opinion-Based		24	30

Note: For Targeted Professional Group and Health Promotion Content Focus, percentages do not add up to 100 because some recommendations targeted more than one group or content focus.

**Table 6 ijerph-17-06249-t006:** Quality appraisal of research articles based on the Mixed Methods Appraisal Tool (MMAT).

Research Design	Number (and Percent) of Articles	Methodological Study Criteria	Total “Yes’s” Across Studies	Total “No’s” Across Studies	Total “Can’t Tell’s” Across Studies
Qualitative	7 (33.3)	Is the qualitative approach appropriate to answer the research question?	7	0	0
		Are the qualitative data collection methods adequate to address the research question?	7	0	0
		Are the findings adequately derived from the data?	6	1	0
		Is the interpretation of results sufficiently substantiated by data?	4	3	0
		Is there coherence between qualitative data sources, collection, analysis and interpretation?	6	1	0
Quantitative–Non-randomized	3 (14.3)	Are the participants representative of the target population?	2	1	0
		Are measurements appropriate regarding both the outcome and intervention (or exposure)?	2	1	0
		Are there complete outcome data?	1	2	0
		Are the confounders accounted for in the design and analysis?	2	1	0
		During the study period, is the intervention administered (or exposure occurred) as intended?	2	0	1
Quantitative–Descriptive	5 (23.8)	Is the sampling strategy relevant to address the research question?	5	0	0
		Is the sample representative of the target population?	4	1	0
		Are the measurements appropriate?	4	1	0
		Is the risk of nonresponse bias low?	1	3	1
		Is the statistical analysis appropriate to answer the research question?	5	0	0
Mixed Methods	6 (28.6)	Is there an adequate rationale for using a mixed methods design to address the research question?	3	3	0
		Are the different components of the study effectively integrated to answer the research question?	3	3	0
		Are the outputs of the integration of qualitative and quantitative components adequately interpreted?	5	1	0
		Are divergences and inconsistencies between quantitative and qualitative results adequately addressed?	1	5	0
		Do the different components of the study adhere to the quality criteria of each tradition of the methods involved?	1	5	0

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
