# Peer review of "Recommendations for Administrators’ Involvement in School-Based Health Promotion: A Scoping Review"

_ijerph, 2020, doi:10.3390/ijerph17176249_

Round 1
Reviewer 1 Report
In general, it is a well written document and is organized according to the journal format.
My main concern with this work is that, in my opinion, it does not have all the information necessary to be repeated by other researchers, something that is very important in a systematic literature review. There are many decisions made in this review that do not have a clear justification.
I have other concerns about the manuscript:
1.- Even though the authors declare to follow the PRISMA statement, to the best of my knowledge, they are not using it properly. Some examples of the above, and taking into account the checklist items (ranging from (1) to (27)) provided by Liberati et al. in [10], are the following:
a) The abstract has not all the elements mentioned in item (2).
b) Research questions are missing in this work (item (3)).
c) PICOS is missing in this work (item (4) and (6)).
d) Year of beginning and end for searching the databases is missing (item (8)).
e) Search string is missing (item (8)).
2.- Introduction.
a) The authors do not explain well why it is important and necessary to do this systematic review, that is, the rational.
b) The research questions to be answered with the systematic review have not been made explicit.
3.- Methods.
a) The PICOS strategy was not used in this work. This can be used even if there are no patients or population involved.
b) It is difficult to understand why the authors used any kind of bibliographic material, without considering the level of validation of them by the scientific community. In general, the systematic literature reviews work with peer-reviewed material. This is to guarantee a minimum level of quality of the results, discussions and conclusions. Otherwise, working with non-validated material could lead to wrong results, discussions and conclusions.
c) In the inclusion/exclusion criteria the authors used "and/or". It is not clear why.
d) The rational behind the inclusion/exclusion criteria is not clear.
e) In section 2.2 the authors said "we sorted the returned records by relevancy and retained the first 100 results from each search". What "relevance" are the authors referring to? Why only the first 100 results?
f) How did the authors determine which were the relevant terms and how did they deduce the dimensions?
g) The authors said "we systematically omitted terms that yielded no relevant results". Then, did they or did they not use the inclusion/exclusion criteria?
h) For example, in dimension 1, when the keyword has more than 2 words, quotation marks are used ("School leader" instead school leader).
i) In section 2.3, why did the authors apply the inclusion/exclusion criteria in the title and abstract separately? Although I have seen that it is done in some reviews, this limits the number of articles to review, with the consequent loss in the quality of the results. In my experience, it is better to apply the criteria in title and abstract together.
4.- Results.
a) It is very difficult to read the results. In general, it is common to see in this part of a systematic review charts and tables, which help to summarize and present the information collected to the readers.
b) The results are not well explained (they look as a report). There is a lack of context.
5.- Discussion.
a) There is not much discussion of the results. It looks like an extension of them. Here it is necessary to discuss the results comparing them with several other previous studies in the area. There are 0.75 references per paragraph.
b) Limitations of this study are missing.
Author Response
In general, it is a well written document and is organized according to the journal format.
Thank you for your encouraging comment.
My main concern with this work is that, in my opinion, it does not have all the information necessary to be repeated by other researchers, something that is very important in a systematic literature review. There are many decisions made in this review that do not have a clear justification.
We hope our extensive revisions, which include a great deal more detail regarding the rationale for the review, the research questions, and methods, address this concern.
I have other concerns about the manuscript:
1.- Even though the authors declare to follow the PRISMA statement, to the best of my knowledge, they are not using it properly. Some examples of the above, and taking into account the checklist items (ranging from (1) to (27)) provided by Liberati et al. in [10], are the following:
a) The abstract has not all the elements mentioned in item (2).
b) Research questions are missing in this work (item (3)).
c) PICOS is missing in this work (item (4) and (6)).
d) Year of beginning and end for searching the databases is missing (item (8)).
e) Search string is missing (item (8)).
Thank you for your careful attention to these details. Based on the suggestion made by the editor, we have chosen to reframe the review as a scoping review and we now state in the manuscript that our review was informed by the PRISMA-ScR guidelines. Where appropriate, we have made sure to also include the details suggested by these guidelines.
2.- Introduction.
a) The authors do not explain well why it is important and necessary to do this systematic review, that is, the rational.
We have added a rationale for conducting the scoping review
b) The research questions to be answered with the systematic review have not been made explicit.
We have stated our research questions.
3.- Methods.
a) The PICOS strategy was not used in this work. This can be used even if there are no patients or population involved.
For scoping reviews, Ghalibaf et al. (2017) suggest using PCC, which we followed in our revisions.
b) It is difficult to understand why the authors used any kind of bibliographic material, without considering the level of validation of them by the scientific community. In general, the systematic literature reviews work with peer-reviewed material. This is to guarantee a minimum level of quality of the results, discussions and conclusions. Otherwise, working with non-validated material could lead to wrong results, discussions and conclusions.
We feel our work is better rationalized now by framing it as a scoping review. Also, as suggested by the editor, we appraised the peer-reviewed research articles that were included in the review using the Mixed Methods Appraisal Tool, and we incorporated the appraisal into our research questions and the discussion.
c) In the inclusion/exclusion criteria the authors used "and/or". It is not clear why.
We agree that this is confusing. We changed and/or to "or" throughout this paragraph.
d) The rational behind the inclusion/exclusion criteria is not clear.
We hope that by adding PCC to this section of the manuscript, our rationale for the inclusion criteria is clearer.
e) In section 2.2 the authors said "we sorted the returned records by relevancy and retained the first 100 results from each search". What "relevance" are the authors referring to? Why only the first 100 results?f) How did the authors determine which were the relevant terms and how did they deduce the dimensions?
g) The authors said "we systematically omitted terms that yielded no relevant results". Then, did they or did they not use the inclusion/exclusion criteria?
We apologize - in regard to e) and g) above, we have reorganized our writing for clarity. We meant to say that we used Google Scholar (sorting results by relevance) to do some preliminary searches of the literature (examining up the first 100 returned records), which helped us determine the most relevant terms and define our dimensions. During this process, we systematically omitted terms that yielded no relevant results.
h) For example, in dimension 1, when the keyword has more than 2 words, quotation marks are used ("School leader" instead school leader).
We have added quotations to our search terms, as suggested.
i) In section 2.3, why did the authors apply the inclusion/exclusion criteria in the title and abstract separately? Although I have seen that it is done in some reviews, this limits the number of articles to review, with the consequent loss in the quality of the results. In my experience, it is better to apply the criteria in title and abstract together.
We appreciate the reviewer's perspective on this matter and agree that this strategy can sometimes serve to unnecessarily eliminate relevant records. However, given the focus of our review, it was very easy in many cases to tell when a returned record was not relevant based on the title. Thus, we feel elimination by titles as a preliminary step is justified for this review.
4.- Results.
a) It is very difficult to read the results. In general, it is common to see in this part of a systematic review charts and tables, which help to summarize and present the information collected to the readers.
Thank you for your suggestion. However, as we have added several other tables to the results in accordance with the research questions we posed, we felt that retaining this portion of the results in a narrative style was preferable.
b) The results are not well explained (they look as a report). There is a lack of context.
We hope that the revisions we made to the results help to fix this issue.
5.- Discussion.
a) There is not much discussion of the results. It looks like an extension of them. Here it is necessary to discuss the results comparing them with several other previous studies in the area. There are 0.75 references per paragraph.
We have expanded our discussion in line with the research questions and added several references.
b) Limitations of this study are missing.
We have added limitations to the conclusions section, as suggested by the editor.
Reviewer 2 Report
Many thanks for the invitation to review this manuscript. The manuscript describes the outcomes of a review to examine recommendations for administrators’ involvement in school-based health promotion. I am very supportive of the concept of this review and feel that it would make a valuable contribution to the field. The paper is very well written, clear and concise making it easy to read. It provides a clear rationale for focusing on the topic area. The methods are clear and appropriate and sufficient detail has been provided for replication. There is a good use of appropriate referencing throughout. Consequently, I recommend that it is accepted for publication. I only have the following comments:
- ‘Fitness’ is repeated as a key word, suggest replacing with an alternative.
- Suggest rewording line 102 of the search strategy as currently it reads as if the search included literature between the periods January to April 2018 when I am assuming that was when the review was conducted?
- This is a suggestion for consideration rather than a request, but I wonder whether Table 2 would benefit from an explanation of each of these themes and/or definition of the categories to make the distinction between them explicit. For example, it is not clear from looking at the table why ‘prioritising PE’ would sit under ‘support’ and not the ‘advocacy’ category.
Author Response
Many thanks for the invitation to review this manuscript. The manuscript describes the outcomes of a review to examine recommendations for administrators’ involvement in school-based health promotion. I am very supportive of the concept of this review and feel that it would make a valuable contribution to the field. The paper is very well written, clear and concise making it easy to read. It provides a clear rationale for focusing on the topic area. The methods are clear and appropriate and sufficient detail has been provided for replication. There is a good use of appropriate referencing throughout. Consequently, I recommend that it is accepted for publication.
Thank you so much for your kind and encouraging remarks!
I only have the following comments:
- ‘Fitness’ is repeated as a key word, suggest replacing with an alternative.
Done.
- Suggest rewording line 102 of the search strategy as currently it reads as if the search included literature between the periods January to April 2018 when I am assuming that was when the review was conducted?
We have revised our phrasing for clarity.
- This is a suggestion for consideration rather than a request, but I wonder whether Table 2 would benefit from an explanation of each of these themes and/or definition of the categories to make the distinction between them explicit. For example, it is not clear from looking at the table why ‘prioritising PE’ would sit under ‘support’ and not the ‘advocacy’ category.
Thank you for your suggestion. Unfortunately, the extra text in the table appeared unwieldy. We hope the narrative portion of the results provides sufficient clarity about each them and subtheme.
Reviewer 3 Report
The article needs to be improved. The Theoretical Framework should be improved and expanded with current bibliographic references and research experiences (minimum between 2015 and 2019). Firstly, there are no specific objectives that structure the analytical discourse and the presentation of the results obtained. The qualitative methodology chosen must be described in a specific way because the authors make a general and simple description of the analysis process. I think that the chosen methodology is the "content analysis" method. In this sense, the authors must explain the phases of the process of content analysis that, for example, Cohen and Manion, Krippendorf, Bardin and/or MacMillan and Schumacher establish to execute the validity and reliability of the process, that is, "transferability" or transference of the results obtained from the process. Finally, in the article it is not clear why the authors decide to make a description of the results having as reference three topics (collaboration, advocacy and support). The structure and description of the results could be through the specific objectives of the research article. In the Conclusion section it is not appropriate to express the objective of the research. This article is considered a first step towards the development of a more complete research, so why does not a statement appear in the "conclusions" section indicating the need to continue this research work? I understand that this article is the basis for future lines of study related to the promotion of healthy schools and what is the type of involvement of administrators.
Author Response
The article needs to be improved. The Theoretical Framework should be improved and expanded with current bibliographic references and research experiences (minimum between 2015 and 2019).
We have improved upon our theoretical and conceptual underpinnings for this review using the Whole School, Whole Community, Whole Child model (2015). This model is considered current and continues to guide much work centered on the intersection of education and public health.
Firstly, there are no specific objectives that structure the analytical discourse and the presentation of the results obtained.
We have refined our purpose statement for clarity and have also stated four clear research questions to guide the review.
The qualitative methodology chosen must be described in a specific way because the authors make a general and simple description of the analysis process. I think that the chosen methodology is the "content analysis" method. In this sense, the authors must explain the phases of the process of content analysis that, for example, Cohen and Manion, Krippendorf, Bardin and/or MacMillan and Schumacher establish to execute the validity and reliability of the process, that is, "transferability" or transference of the results obtained from the process.
After careful review of the literature, we agree that our approach more closely aligns with the traditions of content analysis. We now identify our approach as a qualitative content analysis, drawing from the work of Neuendorf (2017).
Finally, in the article it is not clear why the authors decide to make a description of the results having as reference three topics (collaboration, advocacy and support). The structure and description of the results could be through the specific objectives of the research article.
We have made significant revisions to the results section of the manuscript, which extend beyond the three summarized themes and their subthemes, based on our research questions.
In the Conclusion section it is not appropriate to express the objective of the research. This article is considered a first step towards the development of a more complete research, so why does not a statement appear in the "conclusions" section indicating the need to continue this research work? I understand that this article is the basis for future lines of study related to the promotion of healthy schools and what is the type of involvement of administrators.
We have revised the conclusions section of the manuscript to align with the stated aims and research questions, and we have expressed the importance of our review as an initial step that will hopefully spur increased research into administrator involvement in school-based health promotion.
Round 2
Reviewer 1 Report
I think this work has been significantly improved considering the new approach taken by the authors. I appreciate that the authors have taken my suggestions into account.
I only have a few minor concerns about this work.
1.- In the author's reply, they indicated that they used PCC from Ghalibaf et al. (2017). I think they should add that work to the references.
2.- For the audience, it would be important to see the rational of the search strategy. At this point, I suggest using a table similar to Table 1, page 2 in Breuing et al. 2018, "Barriers and facilitating factors in the prevention of diabetes type II and gestational diabetes in vulnerable groups: protocol for a scoping review", Systematic Reviews, 7:245, DOI:10.1186/s13643-018-0919-y.
3.- Table 1: Only when a search term consists of two or more words is necessary to use quotation marks. Please remove quotation marks in single words.
4.- Table 2:
a) To the best of my knowledge, systematic or similar reviews are not incorporated into a systematic review. This only happens in a systematic review of systematic reviews.
b) Both the total number and the percentage of "Peer-reviewed Article" do not match with the sum of individual document types. Please check both columns.
5.- Above the conclusions in section 5 a line break is missing.
Author Response
I think this work has been significantly improved considering the new approach taken by the authors. I appreciate that the authors have taken my suggestions into account.
Thank you for all your helpful comments.
I only have a few minor concerns about this work.
1.- In the author's reply, they indicated that they used PCC from Ghalibaf et al. (2017). I think they should add that work to the references.
We cited the work that Ghalibaf et al. (2017) relied upon, which is the following:
Peters M, Godfrey C, McInerney P, et al. The Joanna Briggs Institute Reviewers' Manual 2015: Methodology for JBI Scoping Reviews. 2015.
(Please see Line 144.)
2.- For the audience, it would be important to see the rational of the search strategy. At this point, I suggest using a table similar to Table 1, page 2 in Breuing et al. 2018, "Barriers and facilitating factors in the prevention of diabetes type II and gestational diabetes in vulnerable groups: protocol for a scoping review", Systematic Reviews, 7:245, DOI:10.1186/s13643-018-0919-y.
We have added a table in line with the one the reviewer mentioned. (Please see Line 158.)
3.- Table 1: Only when a search term consists of two or more words is necessary to use quotation marks. Please remove quotation marks in single words.
We have made this change. (Please see Line 175.)
4.- Table 2:
a) To the best of my knowledge, systematic or similar reviews are not incorporated into a systematic review. This only happens in a systematic review of systematic reviews.
We agree that systematic reviews of systematic reviews are usually where one would expect to find other systematic reviews included in the records to be reviewed. However, given our purpose was to include all available online documents that contained recommendations for the involvement of school administrators in school-based health promotion, we feel including systematic reviews in our scoping review is warranted.
b) Both the total number and the percentage of "Peer-reviewed Article" do not match with the sum of individual document types. Please check both columns.
Thank you for catching this error. We have corrected it. (Please see Line 326.)
5.- Above the conclusions in section 5 a line break is missing.
We added a line break here. (Please see Line 978.)